

# Evidence for high inter-generational individual quality in yellow-eyed penguins

Aviva M. Stein[1], Melanie J. Young[1], John T. Darby[2], Philip J. Seddon[1] and Yolanda van Heezik[1]

[1] Department of Zoology, University of Otago, Dunedin, New Zealand
[2] Otago Museum, Dunedin, New Zealand

Corresponding author
Yolanda van Heezik,
yolanda.vanheezik@otago.ac.nz

## ABSTRACT

Longitudinal studies focusing on lifetime reproductive success (LRS) have been used to measure individual breeding performance and identify commonalities among successful breeders. By extending the focus to subsequent generations we identify a proportion of high-quality individuals that contribute disproportionately to the population over multiple generations. We used 23 years of yellow-eyed penguin (*Megadyptes antipodes*) breeding data from one breeding area to identify the proportion of individual birds that raised successful breeders, which in turn raised offspring. We explored which life-history components influenced LRS, as this knowledge would enable conservation resources to be focused on high-performing individuals in this endangered population. From 2,147 birds marked as chicks, 370 (17.2%) survived to adulthood and recruited to their natal location, of which 219 (10.2%) fledged offspring: 124 (56.6%) of the 219 birds produced offspring that recruited as breeders. Only 102 birds (4.8% of 2,147) fledged first-generation offspring that in turn fledged offspring (second-generation offspring, or grand-offspring). We found that ~25% of the birds that survived to breed had above-average LRS as well as above-average numbers of grand-offspring, and were more likely to have produced first-generation chicks that recruited and also produced above-average numbers of second-generation chicks. Our findings suggest that there is a core of "super-breeders" that contribute disproportionately to the population over successive generations. Lifespan and age-at-first-breeding were correlated with LRS. We suggest that traits of birds relating to longevity, health (e.g., immunocompetence) and fitness could be examined to identify potential links with high LRS and inter-generational fecundity. "Super-breeders" appear to consistently achieve high LRS and long lifespans in a stochastic environment, demonstrating greater resilience in the face of extreme events.

## INTRODUCTION

Unlike cross-sectional studies across one or two breeding seasons, longitudinal studies based on lifetime reproductive success (LRS) average out occasional breeding failures, and increase the accuracy of measurement of individual success (*Krüger & Lindström, 2001*). The collection of long-term life-history data from a population of marked individuals

makes it possible to identify the proportion of animals that produce recruits, enabling conservation efforts to be efficiently focused on individuals with successful traits (*Moreno, 2003*). Moreover, the overall contribution of individuals with different lifespans or reproductive strategies to subsequent generations can be compared (*Clutton-Brock, 1988*; *Newton, 1989*; *Wooller, Bradley & Croxall, 1992*; *Brommer, Pietiäinen & Kolunen, 1998*).

Studies of LRS have revealed commonalities across bird species: a significant proportion of fledglings from a given population will die before sexual maturity (*Bryant, 1979*; *Newton, 1989*); not all individuals that attempt to breed will be successful; and successful individuals vary in their productivity (*Newton, 1989*). The LRS distribution of a population is typically highly skewed, with large numbers of individuals producing small numbers of young, and only a small proportion of adults producing large numbers of young (*Clutton-Brock, 1988*; *Newton, 1989*). Specific life-history and reproductive traits can be indicative of LRS. Lifespan is the strongest correlate, with longer-lived individuals commonly achieving a higher LRS (*Gustafsson, 1986*; *Clutton-Brock, 1988*; *Newton, 1989*). In seabirds, where individuals can start breeding at various ages, variance in LRS is largely related to variation in breeding lifespan (*Moreno, 2003*), because an increased number of breeding seasons allows individuals more opportunities to successfully fledge offspring.

If only a small proportion of individuals maintain most of the population, the identification of traits affecting LRS of an individual are relevant for conservation efforts. Resources can be diverted towards protecting particularly productive individuals at times when the population is assailed by environmental challenges such as adverse climate conditions, reduced food availability, disease outbreaks or catastrophic events (e.g., oil spills, *Gartrell et al., 2013*). In the case of a pest species, culling efforts could be focused on these highly productive individuals (*Moreno, 2003*).

Factors known to affect reproductive performance in seabirds and which can interact with each other include age, experience, pair bond duration, health condition, sex, number of mates, mate fidelity and site fidelity (*Ryder, 1980*; *Clutton-Brock, 1988*; *Gavin & Bollinger, 1988*; *Bradley et al., 1990*; *Wooller et al., 1990*; *Chastel, Weimerskirch & Jouventin, 1995*). For example, in long-lived seabird species, a period of poor reproductive success at a young age or at a lower level of experience may be superseded by a period at which the individual performs at their peak reproductive output (*Forslund & Pärt, 1995*). At an older age, senescence may begin to reduce reproductive output, followed by terminal illness and death (*Fowler, 1995*; *Nisbet & Dann, 2009*; *Froy et al., 2013*). With increasing age, maternal efficiency might allow for control of the timing, size, volume, composition and pore density of eggs, and high levels of pair synchrony through maintenance of long-term pair bonds may reduce incubation periods and increase nesting success (*Massaro et al., 2002*, *2004*). Breeding skills may therefore improve with both age and experience, as well as with improved synchrony between mated pairs (*Forslund & Pärt, 1995*). The relative importance of each of these factors in estimating LRS is difficult to assess, and can differ dramatically between species and geographic locations.

Breeder quality might not necessarily be age related. Some birds might just be better than others because they have better skills. State-based assessment of individual breeder quality often requires the assumption or prediction that a component of the individual's
health or skill within that system is a driving factor determining its reproductive success (*Wendeln & Becker, 1999*; *Moreno, 2003*), and that this superior skill is independent of age-related performance. These state-based qualities and their relationship with breeding success and LRS can be highly variable within a population of seabirds, but might have low variation for individual birds over time (e.g., common terns *Sterna hirundo*, *Wendeln & Becker, 1999*). Breeding seabirds have a range of responses to catastrophic climatic or weather events, e.g., El Niño Southern Oscillation (ENSO) (*Boersma, 1978*), and vary in the rate at which food is delivered to chicks (*Ens et al., 1992*). Territoriality also can be associated with: individual quality in relation to nesting density (*Stokes & Boersma, 2000*); nest site characteristics (*Stokes & Boersma, 1998*); and rank dominance (*Schubert et al., 2007*).

Lifetime reproductive success measures the number of offspring produced over a lifetime, however it does not consider the viability of those offspring. There may be variability in the quality of offspring produced by different individuals that further reduces the proportion of individuals contributing to subsequent generations. By using a 23-year dataset of individually tracked yellow-eyed penguins we were able to follow reproductive success over more than one generation and identify the proportion of a penguin population that produces grand-offspring. Yellow-eyed penguins (hōiho, *Megadyptes antipodes*) are endemic to New Zealand and listed as "endangered" on the IUCN Red List (*BirdLife International, 2015*). Some mainland populations are intensively managed to mitigate threats posed by introduced predators, disturbance and habitat destruction (*McKinlay, 2001*). Because yellow-eyed penguins are sedentary (*Seddon, van Heezik & Ellenberg, 2013*), long-lived, have high natal philopatry, high breeding site fidelity once breeding and are monogamous (*Richdale, 1957*), they are an ideal species to study LRS. We investigated: (1) the proportion of birds that survive to adulthood; (2) the proportion of adults that breed; (3) the proportion of breeders that produce young that recruit to the breeding population (first-generation); and (4) the proportion of adults producing grand-offspring (second-generation). We also explored the characteristics of highly successful breeders, and the relationship of this trait between generations. We predicted that in line with other seabird species, only a small proportion of yellow-eyed penguins would survive to adulthood, breed, and produce young, and that differences in LRS between males and females exist due to the difference in age-at-first-breeding, with females known to begin breeding earlier than males (*Richdale, 1957*; *Darby & Seddon, 1990*). We predicted that lifespan would have the greatest influence on the number of offspring produced, and be positively correlated with LRS. Yellow-eyed penguins with earlier age-at-first-breeding and fewer overall mates were predicted to have greater LRS.

## METHODS

Yellow-eyed penguins are solitary breeders, in contrast with most other penguin species and seabirds that breed colonially (*Richdale, 1957*; *Darby & Seddon, 1990*). From September to October clutches of up to two eggs are laid, and chicks fledge at ~106 days from late January to late February (*Richdale, 1957*; *Seddon & Davis, 1989*; *Darby & Seddon, 1990*).

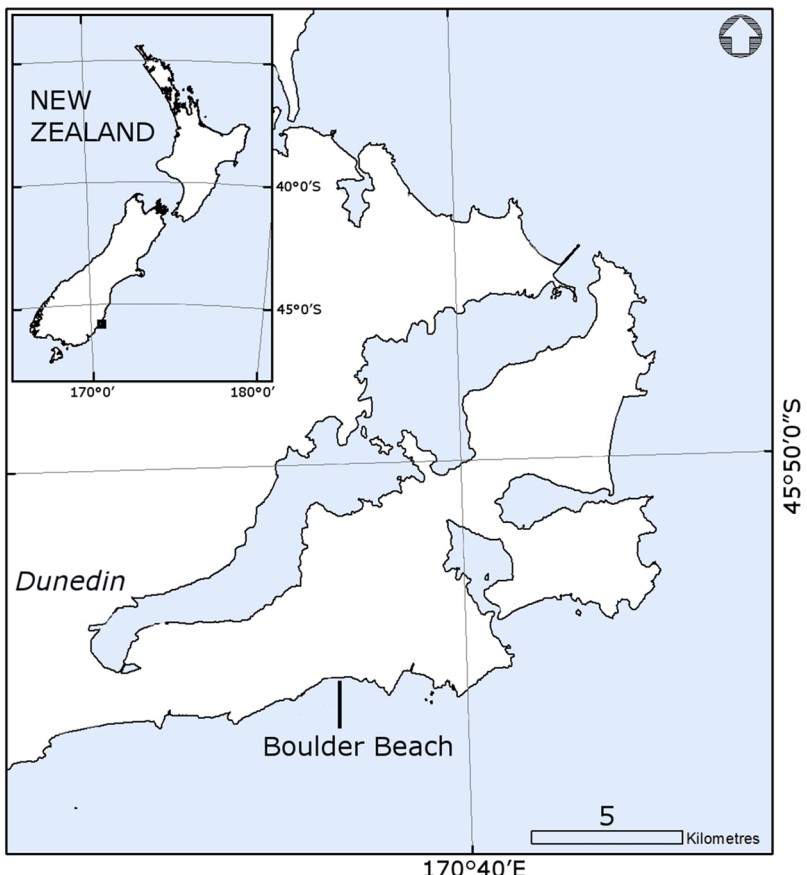

**Figure 1** Map showing the location of Boulder Beach on the Otago Peninsula, Dunedin, New Zealand.

## Yellow-eyed penguin database

We obtained breeding data from the yellow-eyed penguin database administered by the New Zealand Department of Conservation (DOC) and accessed through a Memorandum of Understanding between DOC, representing the contributors to the database, and the University of Otago. We analysed only data from yellow-eyed penguins breeding at the Boulder Beach complex on the Otago Peninsula, New Zealand (45°50′0″S and 170°36′0″E; Fig. 1) because it supports a relatively large population of yellow-eyed penguins, has an inter-decadal history of intensive monitoring and it has been trapped for introduced predators over time. This site has the longest history of chick marking, with the majority of chicks fledged at this site marked with a stainless steel flipper band issued by the New Zealand National Bird Banding Scheme (NZBBS). We acknowledge that the use of flipper bands might present bias (*sensu Petersen et al., 2005*), however in contrast to the foraging ranges of penguin species for which negative impacts of flipper bands have been identified, yellow-eyed penguins are inshore foragers and have much shorter foraging trips (*Seddon & van Heezik, 1990*; *Mattern et al., 2007*). Consequently impacts of flipper bands are likely to be minimal. While negative impacts have been documented for some penguin species (e.g., king penguins, *Aptenodytes patagonicus*, Gauthier-Clerc et al., 2004; Adélie penguins, *Pygoscelis adeliae*, Dugger et al., 2006; little penguins, *Eudyptula minor*,

Hoskins et al., 2008), impacts have been negligible for others (e.g., African penguins, *Spheniscus demersus*, *Barham et al., 2008*; *Hampton, Ryan & Underhill, 2009*; magellanic penguins, *S. magellanicus*, *Boersma & Rebstock, 2009*). A separate study analysing the impact of research manipulations found that even a double banding study on yellow-eyed penguins in one season had no effect on productivity or subsequent survival (*Stein et al., 2017*). Before commencing this study, we completed a comprehensive error check, which involved checking the original notebook records against electronic database records to ensure a high level of accuracy and consistency.

## Data for survival to adulthood and breeding

A total of 2,147 birds were marked at Boulder Beach as chicks or juveniles between 1981 and 2003. We used this sample to calculate the proportion of birds that survived post-fledging, to adulthood (defined as reaching two years of age, when sexually mature), the proportion that recruited to Boulder Beach and attempted to breed, fledged offspring, fledged offspring that survived to adulthood and recruited to the breeding population, and fledged offspring (first-generation) that in turn successfully bred and produced offspring (second-generation or grand-chicks). Birds from cohorts 1981 to 2003 that were still alive or had been sighted after 2007 were excluded from the sample as their breeding lifetime had not ended ($n = 73$ birds).

## Data for LRS analyses

For the detailed analysis of life-history and LRS we used a subset of the data which included information on 130 "founding generation" birds of known sex that survived to breed at least once, from the 1981 to 2003 cohorts, as we considered sex to be an important factor, but sexing information was not always available. We excluded birds whose parents were included in the data subset to avoid pseudoreplication ($n = 87$ birds). We chose the year 2003 as the cut-off cohort, since mean age at first breeding is between three and four years (*Richdale, 1957*): this allowed for birds to have complete lifespans by age 4 in 2007, breed, have offspring that survived to breed (up to 2011) and grand offspring that survived up until the 2014/15 breeding season, when these data were last updated. We did not include birds beyond the 2003 cohort or birds that were alive after 2007, so that we were sure that all of their surviving offspring had recruited to breed by 2014. The latest year a bird with a complete life-history could be recorded as breeding at least once, and be presumed dead was 2007 (i.e., birds from the 2003 cohort would have had a complete lifespan of four years to meet the study's parameters). We excluded birds if they were marked before the 1981 cohort when intensive monitoring began.

We defined LRS as the number of offspring produced by an individual over their lifetime. Only data recorded from birds marked as chicks or as juveniles (1-year old) could be used to ensure complete life-histories, as they were of known age due to plumage differences (duller colouration and lack of the distinctive yellow eye and crown of the adults). Yellow-eyed penguins are typically marked at ~80–90 days, before fledging at ~106 days (*Seddon, van Heezik & Ellenberg, 2013*). If a bird was marked as a juvenile,

we estimated the cohort year by subtracting one year. A bird was considered to have attempted to breed if it or its mate laid an egg.

Analysis of LRS requires complete detectability of the focal population: this would be compromised if individuals bred elsewhere or skipped breeding years and were consequently recorded as having died. We are confident that we have full records of birds due to the intensive monitoring at the study site and annual monitoring at adjacent sites along the coast, and the high level of natal (~81%) and breeding philopatry (~98%), and monogamy exhibited by this species (*Richdale, 1957*; *Ratz et al., 2004*). Birds that skip breeding remain largely undetectable during the breeding season, with only ~8% of skipped birds in our sample being re-sighted as a non-breeder, however detection of breeders is close to 100% (*Hegg et al., 2012*). In our sample, 53 yellow-eyed penguins that survived to breed skipped at least one breeding season once they had established breeding, which is not uncommon, particularly in the year following a poor season, death of a mate or a divorce (*Moore, 1994*; *Ratz et al., 2004*; *Setiawan et al., 2005*). Due to the small, discrete size of nesting areas, the intensity of monitoring at this and in surrounding sites and the high degree of breeding site fidelity, we assumed that if a bird or breeding pair were not seen during multiple visits to the breeding area and to other surrounding areas from early incubation to the end of the guard period they were undertaking a breeding skip. None of the birds we assumed to be undertaking a breeding skip were re-sighted at adjacent monitored breeding areas during their skipped year.

## Sample parameters

The sample parameters for modelling life-history included sex, total number of breeding attempts, recruitment of first-generation offspring, recruitment of successful first-generation offspring, age-at-first-breeding, breeding lifespan, total number of mates and lifespan. We did not include "Cohort" as a factor because the longer-lived birds in later cohorts had not completed their breeding lifetimes, unlike shorter-lived birds. Therefore, including cohort would give a false impression that the super breeder phenomenon ceased at 1994. To ensure that cohort was not an important variable we ran a modified model on a subset of the total data set—the birds from cohorts 1981 to 1994, which included both short-lived and long-lived birds for which we had full LRS ($n = 161$), and found no strong effects (Supplementary Material S2, T2).

Birds were sexed by adult head and foot measurements according to *Setiawan, Darby & Lambert (2004)*. In instances where birds had never been measured or when fledgling measurements were analysed, we inferred the sex from mates where possible, on the assumption that pairings were between males and females only and that the mate had been correctly identified. If there were no measurements or sex recorded for mates, we removed these birds from any lifetime data analysis. We limited our sample for the analysis of life-history traits affecting LRS to birds that survived to breed from the original sample of 2,147 birds, that we were able to sex, and we excluded their offspring ("founding generation," females $n = 62$ and males $n = 68$), so as not to pseudoreplicate breeding pairs or parents and their offspring.

We measured lifespan in whole years at the time of marking as chicks (~3 months old), to the time of either being found dead or "missing" after three consecutive years. Age-at-first-breeding was recorded as the age of the bird during its first recorded breeding attempt. We calculated the number of mates as the minimum possible number of mates, due to 60 of 130 birds in the sample having unidentified mates in some years. We assumed that if a bird's mate was not recorded but it was breeding with a particular bird in the previous and subsequent years that it was the same mate in all three seasons.

## Statistical analysis

We carried out all statistical analyses using R (Version 3.3.1; *R Core Team, 2016*). We used two-sample Wilcoxon rank-sum tests to test for statistical significance between males and females, for parameters including LRS, age-at-first-breeding, recruitment of first-generation offspring, recruitment of successful first-generation offspring, lifespan, total number of mates, total number of breeding attempts and breeding lifespan (see Supplementary Material S1).

The relationships between recruitment of first-generation breeders, successful first-generation breeders, sex and the effect of life-history traits (age-at-first-breeding, lifespan, total number of mates, total number of breeding attempts and breeding lifespan) on LRS were analysed using a generalised linear mixed models (GLMM) with a Poisson distribution and a random factor, mate code, to account for pseudoreplication associated with mated pairs being included in the analysis, using the *lme4* package (*Bates et al., 2015*). Fit of the maximal model was assessed using $R^2 GLMM$ from the *AICcmodavg* package (*Mazerolle, 2016*). We included only uncorrelated variables within the same model ($r < 0.6$; *Hosmer, Lemeshow & Sturdivant, 2013*) to avoid multi-collinearity. We used an information–theoretic approach to model selection, by constructing a maximal model containing all probable input variables (based on a priori reasoning), and then ranking this model against all of its derivatives using QAICc. To account for model selection uncertainty, model-averaging was conducted for the best models ($2\Delta$QAICc, see Supplementary Material S3), using the *MuMIn* and *AICcmodavg* packages in R (*Bartoń, 2016*; *Mazerolle, 2016*).

In order to compare specifically the life-history characteristics (age-at-first-breeding, lifespan, total number of mates, total number of breeding attempts and breeding lifespan) between birds which proved over the two generations to be highly successful breeders ("high-quality") and the remainder of the birds ("ordinary"), we defined the highly successful individuals as follows: those birds with above-average LRS relative to their sex (females $\geq 7$, males $\geq 6$) and those that had an above-average number of grand-offspring (second-generation chicks; females $\geq 10$, males $\geq 6$). Using the same GLMM approach, we analysed the effect of life-history parameters on the LRS of these two groups of birds to determine differences in breeder quality.

## RESULTS

Of the total sample of 2,147 birds marked as chicks or juveniles from 1981 to 2003, 1,546 (72.0%) were thought to have died before reaching adulthood, whereas 441 birds survived

**Table 1** Breeding and recruitment overview of numbers and percentages of individual yellow-eyed penguins marked between 1981 and 2003 at Boulder Beach, Otago Peninsula, New Zealand ($n = 2147$).

|  | Number | Percent |
| --- | --- | --- |
| Marked as chick or juvenile at Boulder Beach | 2,147 | |
| Marked chicks that were never re-sighted | 1,546 | 72.0 |
| Marked chicks that were re-sighted under two years | 601 | 28.0 |
| Survived to adulthood (two years) | 441 | 20.5 |
| *Sighted at Boulder Beach* | *370* | *17.2* |
| *Sighted elsewhere* | *71* | *3.3* |
| Attempted breeding at Boulder Beach | 264 | 12.3 |
| Fledged offspring at Boulder Beach | 219 | 10.2 |
| Fledged first-generation offspring that recruited | 124 | 5.8 |
| Fledged successful first-generation offspring | 102 | 4.8 |

to be seen at least once as an adult: 71 of these birds were sighted away from Boulder Beach at other monitoring locations where they subsequently bred. Of the 370 birds that were re-sighted at Boulder Beach as an adult at least once (17.2%), 264 attempted to breed at least once (12.3%) and 219 bred successfully at least once (10.2%). Only 124 birds produced at least one first-generation chick that recruited to the breeding population and attempted to breed at least once (5.8%), and 102 had first-generation offspring that not only recruited but bred successfully at least once (4.8%). Overall figures are presented in Table 1.

## LRS of male and female yellow-eyed penguins

There was high individual variance in LRS calculated for both males and females ($n = 130$), with this variance being higher for females (Table 2; Fig. 2). The maximum number of total offspring a female yellow-eyed penguin produced was 24, compared to 23 for males. The only significant differences between males and females were for recruitment of breeders ($W = 2,554$, $p = 0.03$) and recruitment of successful breeders ($W = 2,615$, $p = 0.01$). There was no difference between males and females for LRS ($W = 2,452$, $p = 0.11$), age-at-first-breeding ($W = 1750.5$, $p = 0.08$), lifespan ($W = 2169.5$, $p = 0.77$), total number of mates ($W = 2187.5$, $p = 0.69$), total number of breeding attempts ($W = 2370$, $p = 0.22$) and breeding lifespan ($W = 2359.5$, $p = 0.24$).

## Relationships between fledging and recruitment

There was a strong positive relationship between number of chicks fledged per parent (LRS) and number that recruited for females ($\lambda(\text{Female}_i) = \exp(-0.27) \times \exp(0.12 \times \text{LRS}_i)$) and males ($\lambda(\text{Male}_i) = \exp(-0.60) \times \exp(0.13 \times \text{LRS})$). There was also a significant positive relationship between the number of chicks fledged (LRS) and number of successful recruits (i.e., recruits that in turn successfully fledged offspring during at least one breeding attempt) for females ($\lambda(\text{Female}_i) = \exp(-0.65) \times \exp(0.12 \times \text{LRS}_i)$) and males ($\lambda(\text{Male}_i) = \exp(-1.26) \times \exp(0.14 \times \text{LRS})$).

**Table 2** Mean LRS, number of recruits, number of recruits that bred successfully, lifespan, age-at-first-breeding, breeding lifespan, number of breeding attempts and number of mates of founding generation female ($n = 62$) and male ($n = 68$) yellow-eyed penguins breeding at Boulder Beach, New Zealand.

| Variable | Mean | Var | SE | Min | Med | Max |
|---|---|---|---|---|---|---|
| | *Females* | | | | | |
| LRS | 6.82 | 33.8 | 0.74 | 0 | 6 | 24 |
| Recruits | 2.24 | 5.01 | 0.28 | 0 | 2 | 9 |
| Successful recruits | 1.52 | 2.71 | 0.21 | 0 | 1 | 7 |
| Lifespan (years) | 9.44 | 32.77 | 0.73 | 2 | 7.5 | 24 |
| Age at first breeding | 3.60 | 3.16 | 0.23 | 2 | 3 | 12 |
| Breeding lifespan (years) | 5.84 | 26.69 | 0.66 | 0 | 4 | 17 |
| Breeding attempts | 5.79 | 17.74 | 0.53 | 1 | 4.5 | 16 |
| Total mates | 2.03 | 1.57 | 0.16 | 1 | 2 | 7 |
| | *Males* | | | | | |
| LRS | 5.07 | 22.07 | 0.57 | 0 | 4 | 23 |
| Recruits | 1.38 | 2.81 | 0.20 | 0 | 1 | 6 |
| Successful recruits | 0.81 | 1.29 | 0.14 | 0 | 0 | 5 |
| Lifespan (years) | 8.87 | 25.3 | 0.61 | 2 | 8 | 21 |
| Age at first breeding | 4.09 | 3.48 | 0.23 | 2 | 3 | 11 |
| Breeding lifespan (years) | 4.78 | 21.7 | 0.56 | 0 | 3 | 18 |
| Breeding attempts | 4.78 | 11.60 | 0.41 | 1 | 4 | 14 |
| Total mates | 1.93 | 1.32 | 0.14 | 1 | 2 | 6 |

**Notes:**
Var, variance; SE, standard error; Min, minimum; Med, median; Max, maximum.

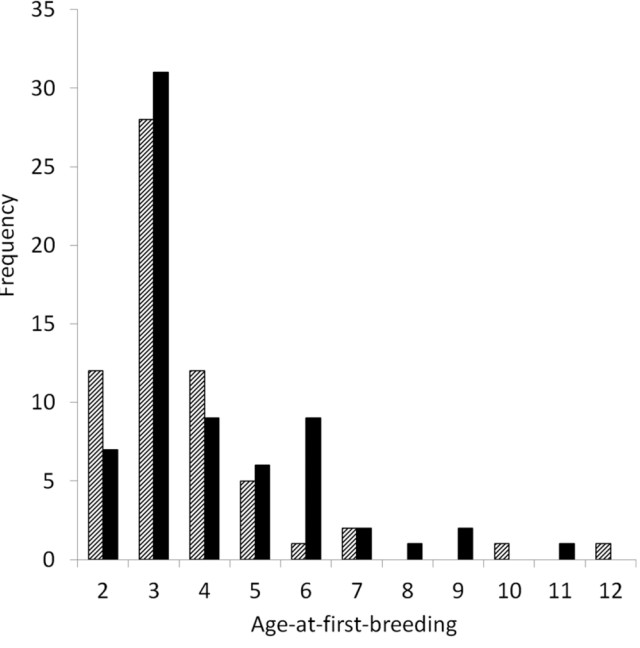

**Figure 2 Age-at-first-breeding (years) of female (striped, $n = 62$) and male (black, $n = 68$) yellow-eyed penguins with complete life histories that bred at Boulder Beach, Dunedin, New Zealand.**

**Table 3** Model-averaged generalised linear mixed effects model of lifetime reproductive success (LRS) and life-history parameters of 130 founding generation yellow-eyed penguins ($n$ = 809 breeding attempts) that were marked at Boulder Beach, New Zealand between 1981 and 2003.

| Coefficients | Estimate | SE | 95% confidence interval | Relative importance |
|---|---|---|---|---|
| Intercept[1] | 2.02 | 0.04 | 1.94, 2.09 | – |
| Sex (male)[2] | −0.16 | 0.04 | −0.25, −0.08 | 1.00 |
| $z$ (age at first breeding)[2] | −0.14 | 0.04 | −0.21, −0.07 | 1.00 |
| $z$ (lifespan)[2] | 0.61 | 0.03 | 0.55, 0.68 | 1.00 |
| Sex (male): $z$ (lifespan) | 0.01 | 0.03 | −0.07, 0.13 | 0.24 |
| $z$ (total mates) | −0.004 | 0.02 | −0.08, 0.04 | 0.23 |

**Notes:**
[1] Sex (female) is the reference category.
[2] Significant results.
Model statement: glmer(LRS ~ $z$(life span) + $z$(age at first breeding) + $z$(total mates) + factor(sex) + factor(sex):$z$(age at first breeding) + factor(sex):$z$(life span) + (1|mate code).
All non-binary data are standardised to have mean = 0 and SD = 1.

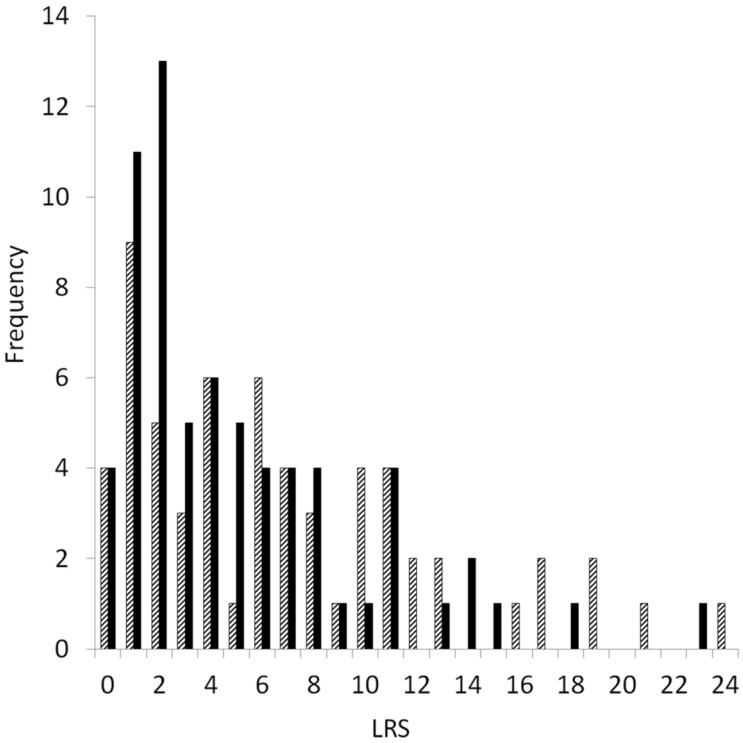

**Figure 3** Frequencies of the total number of chicks fledged (lifetime reproductive success, LRS) by female (striped, $n$ = 62) and male (black, $n$ = 68) yellow-eyed penguins with complete life histories that bred at Boulder Beach, New Zealand.

## Life-history traits

Lifespan was the strongest positive correlate of LRS (0.61 ± 0.03), followed by a negative correlation with age-at-first-breeding (−0.14 ± 0.04; GLMM $R^2$m = 0.56, $R^2$c = 0.79; Table 3). There was a trend associated with sex, with males having slightly lower LRS compared to females (Fig. 3). There was no association with the number of mates and LRS; or interactions between sex, age-at-first-breeding or lifespan (Table 3).
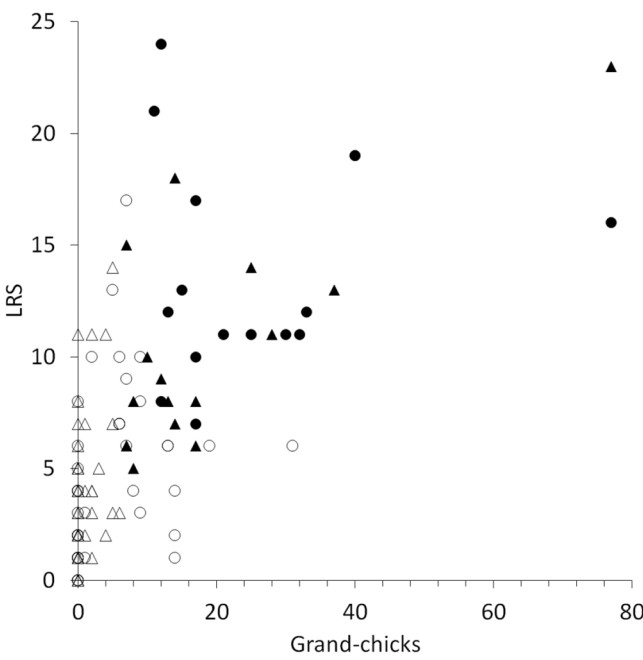

**Figure 4** The relationship between lifetime reproductive success (LRS) and the production of second-generation offspring (grand-offspring) for female (filled dots = "high-quality breeder," empty dots = "low-quality breeder") and male (filled triangles = "high-quality breeder" and open triangles = "low-quality breeder") yellow-eyed penguins with complete life histories that bred at Boulder Beach, New Zealand.

## Determining the traits of "high-quality" breeders

"High-quality" birds (i.e., those with above-average LRS and higher than average numbers of successful breeding offspring; females = 16, males = 16) produced 389 fledged chicks, of which 133 first-generation offspring recruited and 94 were successful, producing 713 grand-offspring (second-generation chicks) (Fig. 4). These higher-quality breeding birds produced 26 first-generation offspring with an above-average LRS $\geq$ 6. In contrast, the remaining "ordinary" birds (females = 46, males = 52) produced 379 chicks, of which 100 first-generation offspring recruited, and 55 were successful in producing 252 grand-offspring (second-generation chicks) (Fig. 4). The ordinary breeding birds ($n$ = 98) produced 10 above-average first-generation offspring.

An interaction effect between breeder type and lifespan was detected, but there was no interaction between age-at-first breeding and breeder type (Table 4), despite the ordinary female birds beginning breeding at least one year younger than the high-quality female birds (Table 5). The ordinary birds had shorter lifespans and therefore lower LRS, with high-quality birds having on average double the longevity of their short-lived conspecifics (Table 5).

## DISCUSSION

By tracking reproductive success in yellow-eyed penguins over more than one generation we show that only a small proportion of fledglings survive, recruit and attempt to breed, however LRS and survival appears to be an inter-generational trait, with above-average

**Table 4 Model-averaged generalised linear mixed-effects model of lifetime reproductive success in yellow-eyed penguins marked at Boulder Beach between 1981 and 2003, including breeder quality as well as.**

| Coefficients | Estimate | SE | 95% confidence interval | Relative importance |
|---|---|---|---|---|
| Intercept[1] | 1.84 | 0.04 | 1.77, 1.91 | – |
| Breeder quality (higher-quality)[2] | 0.40 | 0.06 | 0.29, 0.51 | 1.00 |
| $z$ (age at first breeding)[2] | −0.10 | 0.03 | −0.15, −0.05 | 1.00 |
| $z$ (lifespan)[2] | 0.60 | 0.04 | 0.52, 0.67 | 1.00 |
| Breeder quality (higher-quality): $z$ (lifespan)[2] | −0.21 | 0.05 | −0.31, −0.11 | 1.00 |
| $z$ (total mates) | 0.01 | 0.02 | −0.03, 0.08 | 0.34 |

**Notes:**
[1] Breeder quality (lower-quality) is the reference category.
[2] Significant results.
Model statement: glmer(LRS ~ $z$(life span) + $z$(age at first breeding) + $z$(total mates) + factor(breeder type) + factor(breeder type):$z$(age at first breeding) + factor(breeder type):$z$(life span) + (1|mate code).
All non-binary data are standardised to have mean = 0 and SD = 1.

breeders more likely to produce chicks that will be highly successful breeders. Less than 1.5% (32 of 2147) of these breeding birds are what we call "super-breeders." These "super-breeders" appear to be successful in producing offspring that will themselves survive and go on to contribute disproportionately to the next generation.

Only 10.2% of the sample population of 2,147 fledgling yellow-eyed penguins eventually recruited and produced offspring at all, meaning that 89.8% of young fledged did not contribute to the next generation at Boulder Beach. Low juvenile survival is likely to be the principal reason for the low number of penguins recruiting to breeding populations. Only 20.5% of yellow-eyed penguin fledglings survived to sexual maturity (two years of age, including birds that did not recruit), a similar proportion to the 20.8% yellow-eyed penguins re-sighted as sexually mature adults at the Boulder Beach complex between 1981 and 1990 (*Efford, Darby & Spencer, 1996*), although our reported result includes 3.3% of birds that were re-sighted away from Boulder Beach. Our survival to adulthood rate was higher than the 10.4% of Adélie penguins (*P. adeliae*) that survived to age two (*Ainley & DeMaster, 1980*), but low compared to the range of values for survivorship from fledging to sexual maturity for 19 species of passerines and seabirds (42–86%; *Newton, 1989*). Survivorship to two years was significantly lower than reported in several other studies of seabirds, including 57.6% for common guillemots (*Uria aalge*; *Crespin et al., 2006*), 41–54% for sooty shearwaters (*Puffinus griseus*; *Fletcher et al., 2013*), and ~77% for king penguins after 1 year (*A. patagonicus*; *Saraux et al., 2011*). It was even lower than the ~32% of yellow-eyed penguins that survived to age two between 1936 and 1952 (*Richdale, 1957*).

The probability that birds survive the period between parental care and adulthood has a large influence on population dynamics, but is highly variable (*Maness & Anderson, 2013*). The most common hypothesis for high rates of mortality in young birds is their lack of experience, poor foraging skills and physical immaturity (*Lack, 1954*; *Ashmole, 1963*; *Orians, 1969*; *Dunn, 1972*). Positive correlations are predicted between body mass and juvenile survival, based on the assumption that heavier juveniles have fat reserves that buffer the food limitation associated with inexperience (*Lack, 1966*; *McClung et al., 2004*;

**Table 5** Mean LRS, number of first-generation offspring, number of first-generation offspring that bred successfully, lifespan, age-at-first-breeding, breeding lifespan, number of breeding attempts and number of mates of ordinary and high-quality yellow-eyed penguins breeding at Boulder Beach, New Zealand.

| Variable | Mean | Var | SE | Min | Med | Max |
|---|---|---|---|---|---|---|
| *Females (ordinary breeders, n = 46)* | | | | | | |
| LRS | 4.37 | 14.06 | 0.55 | 0 | 4 | 17 |
| Recruits | 1.33 | 2.27 | 0.22 | 0 | 1 | 5 |
| Successful recruits | 0.83 | 1.12 | 0.16 | 0 | 0 | 4 |
| Lifespan (years) | 7.48 | 23.63 | 0.72 | 2 | 6 | 22 |
| Age at first breeding | 3.30 | 2.57 | 0.24 | 2 | 3 | 12 |
| Breeding lifespan (years) | 4.17 | 20.46 | 0.67 | 0 | 3 | 17 |
| Breeding attempts | 4.15 | 9.51 | 0.45 | 1 | 3.5 | 13 |
| Total mates | 1.80 | 1.36 | 0.17 | 1 | 1 | 7 |
| *Females (high-quality breeders, n = 16)* | | | | | | |
| LRS | 13.88 | 23.85 | 1.22 | 7 | 12 | 24 |
| Recruits | 4.88 | 3.58 | 0.47 | 2 | 4.5 | 9 |
| Successful recruits | 3.5 | 2 | 0.35 | 1 | 3 | 7 |
| Lifespan (years) | 15.06 | 16.86 | 1.03 | 7 | 16 | 24 |
| Age at first breeding | 4.44 | 4.13 | 0.51 | 3 | 4 | 10 |
| Breeding lifespan (years) | 10.63 | 14.25 | 0.94 | 4 | 9.5 | 17 |
| Breeding attempts | 10.5 | 11.73 | 0.86 | 5 | 9 | 16 |
| Total mates | 2.69 | 1.70 | 0.33 | 1 | 3 | 5 |
| *Males (ordinary breeders, n = 52)* | | | | | | |
| LRS | 3.42 | 9.82 | 0.43 | 0 | 2 | 14 |
| Recruits | 0.75 | 1.09 | 0.14 | 0 | 0 | 4 |
| Successful recruits | 0.33 | 0.30 | 0.08 | 0 | 0 | 2 |
| Lifespan (years) | 7.58 | 19.31 | 0.61 | 2 | 6 | 19 |
| Age at first breeding | 4.04 | 3.45 | 0.26 | 2 | 3 | 11 |
| Breeding lifespan (years) | 3.54 | 15.43 | 0.54 | 0 | 2.5 | 14 |
| Breeding attempts | 3.87 | 9.06 | 0.42 | 1 | 3 | 12 |
| Total mates | 1.83 | 1.52 | 0.17 | 1 | 1 | 6 |
| *Males (high-quality breeders, n = 16)* | | | | | | |
| LRS | 10.44 | 25.06 | 1.25 | 5 | 8.5 | 23 |
| Recruits | 3.44 | 2.93 | 0.43 | 1 | 3 | 6 |
| Successful recruits | 2.38 | 1.32 | 0.29 | 1 | 2 | 5 |
| Lifespan (years) | 13.06 | 22.6 | 1.19 | 8 | 12 | 21 |
| Age at first breeding | 4.25 | 3.8 | 0.49 | 2 | 3.5 | 9 |
| Breeding lifespan (years) | 8.81 | 21.63 | 1.16 | 3 | 6.5 | 18 |
| Breeding attempts | 7.75 | 8.73 | 0.74 | 4 | 7 | 14 |
| Total mates | 2.25 | 0.6 | 0.19 | 1 | 2 | 4 |

**Notes:**
Var, variance; SE, standard error; Min, minimum; Med, median; Max, maximum. Continued overleaf.

*Maness & Anderson, 2013*). Yellow-eyed penguins are sedentary foragers that lack a long-distance migratory phase in their life-history, however juveniles undergo a pelagic phase lasting for up to two years, during which time they are sighted only erratically along

the coast. No information exists on where juveniles disperse to (*Darby & Seddon, 1990*), and most mortality occurs during this post-fledging pelagic phase. It is unclear whether the low survival of juvenile penguins in this study is normal or depressed by changing environmental conditions.

This study indicates that of the chicks seen again at Boulder Beach after fledging (24.7% of the original sample), 69.8% were seen at two years of age (i.e., up to the onset of sexual maturity, 69.8%, Table 1), but only 49.8% survived to breed at least once. These figures suggest that juvenile mortality occurs in two or more stages: as high post-fledging mortality due to inexperience, immaturity and lack of skill and possibly due to seasonal fluctuations in prey availability later in the breeding season when juveniles must prepare for their first annual moult. The difference in juvenile survival between *Richdale's (1957)* study and ours may be indicative of an adverse change in foraging conditions (*Browne et al., 2011*; *Mattern et al., 2014*), entanglement in recreational or commercial fishing gear (*Darby & Dawson, 2000*), increasing frequency of poor seasons (*van Heezik, 1990*) and competition with or predation by recovering otariid populations (*Bradshaw, Lalas & Thompson, 2000*; *Lalas et al., 2007*), all of which are documented to affect adult yellow-eyed penguins. Marine pollution that results in disease outbreaks and mass mortality events, have also been hypothesised (e.g., 1990 mass mortality event, *Gill & Darby, 1993*; e.g., diphtheritic stomatitis, *Alley et al., 2004*, *2017*; *Trathan et al., 2015*).

From the sample of 2,147 fledglings, the proportions that survived and attempted breeding (12.3%), fledged offspring (10.2%) and fledged offspring that recruited (5.8%) seem low. However, the proportion of birds that attempted breeding and were successful is relatively high (219 of 264, 82.9%). In other words, if a bird was successful in surviving to make a breeding attempt, there was an 83% probability that it would be successful in fledging at least one chick in its lifetime, a 47% (124/264) probability it would fledge at least one chick that would recruit to the breeding population, and a 39% (102/264) probability that the bird would fledge chicks that would recruit and subsequently fledge offspring. The proportion of yellow-eyed penguins surviving to attempt to breed at least once was comparatively lower than in red-billed gulls (*Larus novaehollandiae*, 18–22%), little penguins (*E. minor*, 28–35%), kittiwakes (*Rissa tridactyla*, 34–42%) and short-tailed shearwaters (*Puffinus tenuirostris*, 69–73%). However, the proportion of breeding yellow-eyed penguins that produced recruits (47%) is one of the highest, with only kittiwakes having similar recruitment rates (~41–50%) (*Coulson, 1988*; *Wooller et al., 1988*; *Mills, 1989*; *Dann & Cullen, 1990*; *Moreno, 2003*). In yellow-eyed penguins, recruitment into the breeding population appears to be driven in part by the higher survival rate of the offspring of a subset of breeders, high-quality birds labelled here as "super-breeders," producing more recruits (133 first-generation recruits from 32 birds, 57.1%) than the ordinary breeders (100 first-generation recruits from 98 birds, 42.9%).

Both male and female penguins that survived to breed varied considerably in the total number of offspring they fledged. For a long-lived species, the average number of fledged young seems relatively small (female mean = 6.82, male mean = 5.07), but falls within the range of values reported from the few studies that have estimated mean LRS in seabirds,

**Table 6** Comparison of LRS and maximum number of young fledged by individuals of five different bird species for males and females (where data were available from *Coulson, 1988*; *Mills, 1989*; *Dann & Cullen, 1990*; *Korpimäki, 1992*; *Krüger & Lindström, 2001*; *Garamszegi et al., 2004*).

| Species | LRS (female) | LRS (male) | Max fledged (female) | Max fledged (male) |
|---|---|---|---|---|
| Yellow-eyed penguin (*Megadyptes antipodes*) | 6.82 | 5.07 | 24 | 23 |
| Black-legged kittiwake (*Rissa tridactyla*) | 6.93 | 7.41 | – | – |
| Red-billed gull (*Larus novaehollandiae*) | 3.4 | 3 | 26 | 28 |
| Little penguin (*Eudyptula minor*) | 2.28 | 2.13 | 35 | 44 |
| Tengmalm's owl (*Aegolius funereus*) | – | 5.2 | – | 26 |
| Common buzzard (*Buteo buteo*) | 3.48 | 2.72 | 20 | 20 |
| Collared flycatcher (*Ficedula albicollis*) | 5.18 | – | – | – |

passerines and birds of prey, demonstrating that LRS for many species of birds remain similar as a result of life-history trade-offs (Table 6). Females had slightly longer lifespans and longer breeding lifespans than males, because females started breeding earlier than males. Females may have more opportunities to breed than males, due to an apparent sex-skew, with males outnumbering females (*Richdale, 1957*). The maximum number of fledged offspring for both male (23) and female yellow-eyed penguins (24) was much higher than mean values, reflecting the highly negatively skewed distribution of LRS (Fig. 2). This is consistent with the observation that most individuals produce small numbers of young, and only a few produce many (*Newton, 1989*). Nevertheless, there was a wide range in the number of young fledged by individual birds regardless of sex, despite the greater cost of reproduction incurred by breeding females. *Newton (1989)* concluded that LRS is generally similar for males and females in species that lack high levels of sexual dimorphism, which is the case for yellow-eyed penguins (*Seddon, van Heezik & Ellenberg, 2013*).

## LRS predictors

Lifespan was the strongest correlate of LRS, with the number of offspring produced increasing significantly with increased lifespan. This trend is very common for seabirds (*Clutton-Brock, 1988*; *Newton, 1989*, *1995*), and is attributed to a number of factors: increased breeding opportunities, and increasing parental experience with lifespan, as has been demonstrated in other seabird species (*Limmer & Becker, 2009*; *Saraux et al., 2012*), and the general fitness required for a long lifespan. Long-lived birds are the primary contributors to the gene pool in many species, meaning there is likely to be selection for viability (*Moreno, 2003*; *Mauck, Huntington & Grubb, 2004*). We found that high-quality breeders had lifespans that were on average double that of ordinary breeders, but they produced three to four times more offspring in their lifetimes than did ordinary birds (Table 5). In other species, lifespan explains less of the variance when the number of recruits is examined as opposed to number of offspring produced (*Newton, 1989*).
In contrast, we found a highly significant relationship between LRS and the number of recruits and successful recruits produced for yellow-eyed penguins, meaning that the characteristics of birds with longer lifespans are likely to be reliable predictors of parental quality for this species.

Age-at-first-breeding was the second strongest predictor of LRS in yellow-eyed penguins, with birds that began breeding later having lower lifetime totals of offspring, due to a decrease in total breeding opportunities (*Newton, 1989*). The theory of antagonistic pleiotropy suggests that high early-life reproductive output is at the expense of later-life productivity, and can be selected for if selection is stronger at early stages of life, so that early benefits outweigh later costs (*Williams, 1957*). While there was a difference between the LRS of males and females in this study, no interaction effect could be detected, despite earlier reproduction in females, which may potentially result in accelerated reproductive senescence (*Partridge, 1992*; *Reed et al., 2008*). This trend has been observed in several long-lived bird species, which all showed a positive correlation between age-at-first-breeding and survival in females, suggesting a trade-off between early recruitment and lifespan (*Ollason & Dunnet, 1978*; *Ainley & DeMaster, 1980*; *Pyle et al., 1997*; *Tavecchia et al., 2001*). Individual variation in LRS for yellow-eyed penguins therefore appears to be due to variation in lifespan (1–24 years) and age-at-first-breeding (2–12 years), together determining the length of the breeding lifespan (1–18 years).

It is common for many species of seabirds to show reduced breeding success after changing mates, most likely due to a trade-off in time and energy expenditure for finding a new mate and foraging, and also due to lack of familiarity with the new mate (*Ollason & Dunnet, 1978*; *Coulson Thomas, 1985*; *Newton, 1989*). In short-tailed shearwaters a mate change results in a temporary decrease in breeding success, but this effect is smaller in individuals that are more experienced breeders (*Wooller et al., 1989*). Breeding success of male common guillemots decreased with an increasing number of mates (*Lewis et al., 2006*). Yellow-eyed penguins that change mates are more likely to experience breeding failure the subsequent year (*Setiawan et al., 2005*). We did not detect a significant negative effect of number of mates on lifetime number of offspring produced, possibly due to the tendency for longer-lived birds to outlive their mates, resulting in higher overall numbers of mates.

## Conservation implications

Chronic and acute stress as a result of climate change, marine pollution, disturbance at terrestrial breeding sites and extreme nutritional stress may decrease LRS, as the cumulative effects of increasing types of stressors force individuals to reduce their investment in productivity, increase breeding skip behaviours (e.g., red-footed boobies *Sula sula*, *Cubayanes et al., 2011*) or result in breeder mortality (*Kitaysky et al., 2010*). In black-legged kittiwakes, breeding behaviour is mediated by increased corticosterone production during periods of poor food supply (*Kitaysky et al., 2010*; *Schultner et al., 2013*). Clarifying the factors that separate the success of the few that produce many offspring from the many that do not may therefore need to take into account the role of chronic or acute stress on the parameters that may be used to measure their fitness. Likewise, birds that contribute disproportionately to successive generations may have higher thresholds for anthropogenic and environmental stressors than average birds. The impact of extreme events on different phenotypes of conspecifics may differ as a consequence of the "super-breeder" phenomenon, since these birds tend to consistently achieve high LRS and long-lifespans in a stochastic environment.

It appears that the Boulder Beach population of yellow-eyed penguins is sustained by a small proportion of high-quality, long-lived birds, the "super-breeders." High levels of philopatry may drive high-quality and ordinary breeding recruits to return to their natal area, and once they begin breeding they are likely to remain at these breeding sites for life. This behaviour may be hazardous for population stability should either one of their marine or terrestrial habitats become threatened. If circumstances require it necessary to protect specific individuals in a population from catastrophe or to differentially allocate resources due to budget constraints, it would be important to distinguish between potentially very successful breeders and the evolutionary "living dead" (*Moreno, 2003*). Oiling is the greatest anthropogenic threat to penguins (*Trathan et al., 2015*), requiring triage of breeders for temporary captive management. The unexplained mass mortalities of adult and juvenile yellow-eyed penguins on the Otago Peninsula in 1990, 1996 and 2013 due to exposure to an unknown toxic agent have presented conservation managers with opportunities to safeguard specific individuals from harm (*Gill & Darby, 1993*; DOC, Young MJ, 2013, unpublished data). While effort should be placed on safeguarding all individuals in a threatened population during a period of catastrophe, only a small proportion of individuals will contribute to the recovery of the population following such an event.

Although it seems sensible to focus conservation resources on "super-breeders," the challenge lies in identifying them. The positive relationship between LRS and the number of successful recruits indicates that birds demonstrating relatively high LRS are also those that produce high-quality offspring. Lifespan is the main predictor of LRS, but unfortunately it cannot be calculated until the death of an individual. Age-at-first-breeding can be identified before death, although its association with potential LRS is much weaker, however high-quality female birds tended to recruit a year later than ordinary birds. Fledging weight of the individual could indicate that the individual comes from a high-quality lineage and will be more likely to survive (*McClung et al., 2004*) and go on to breed and produce high-quality offspring that also fledge at a higher than average weight. We could not explore this relationship with our dataset as fledging weights were not reliably recorded in the historical records. The value of other potential indicators of living super-breeders include oxidative stress, white blood cell counts, hue and size of the coloured eye and eye stripe; these are being explored in ongoing research.

It may be possible to single out birds on the basis of life-history traits that relate to state-based quality. The importance of state-based assessments for yellow-eyed penguins has yet to be fully explored, especially with regard to analysing the immunocompetence of individuals. Disease prevalence has increased in recent years (*Alley et al., 2004*, *2017*; *Hill et al., 2010*; *Argilla et al., 2013*). *Moreno et al. (1998)* measured variables related to health state and cell-mediated immunity between early and late breeders for chinstrap penguins (*Pygoscelis antarcticus*), finding that early breeders experienced better health than later breeders. Female chinstrap penguins with leucocytosis laid smaller eggs, had slower chick growth rates and were more prone to failure (*Moreno et al., 1998*). Information about foraging ecology, particularly in young birds is also necessary. Foraging strategies in high-quality Adélie penguins have been linked to better provisioning of

chicks, suggesting that some birds may be physiologically more capable by virtue of genetic superiority (*Lescroël et al., 2010*).

Around half of all seabird species globally are thought to be in decline, many of which have restricted numbers and ranges and demographic characteristics that severely limit their rate of recovery (*Croxall et al., 2012*). As long-lived species, and with demographic characteristics similar to those of yellow-eyed penguins, it is likely that population persistence in other species is also dependent to some extent on a subset of successful breeders. Understanding what causes some birds to display enhanced resilience and identifying and protecting these individuals may be vital in the face of growing threats.

## ACKNOWLEDGEMENTS

This study would not have been possible without the following individuals, who contributed their data: Kerri-Anne Edge, Ursula Ellenberg, Mike Hazel, Melanie Massaro, Thomas Mattern, Dean Nelson and Alvin Setiawan. Thanks also to Bruce McKinlay and Dave Houston (DOC) for access to the Yellow-eyed Penguin Database and David Ainley and two anonymous reviewers whose constructive comments greatly improved the manuscript.

### Funding

The authors received no funding for this work.

### Competing Interests

Yolanda van Heezik is an Academic Editor for PeerJ.

### Author Contributions

- Aviva M. Stein conceived and designed the experiments, analysed the data, wrote the original draft of the paper.
- Melanie J. Young conceived and designed the experiments, analysed the data, wrote the paper, prepared figures and/or tables, reviewed drafts of the paper.
- John T. Darby contributed data, overview.
- Philip J. Seddon conceived and designed the experiments, reviewed drafts of the paper.
- Yolanda van Heezik conceived and designed the experiments, reviewed drafts of the paper.

### Animal Ethics

The following information was supplied relating to ethical approvals (i.e., approving body and any reference numbers):

Animal ethics approval was not necessary as this study involved accessing data from a database, no manipulations were performed.

## Data Deposition

The raw data for all analyses of LRS have been supplied as a Supplementary File, plus a code file.

The figures in the first table on survival to breed cannot be supplied because they are calculated from data in an entire 30-year database, that requires permission via an MOU to access. We have the necessary permission for this study, but can't provide public access to the entire database. We have provided the raw data for the LRS analyses, which are the core of the paper.

## Supplemental Information

Supplemental information for this article can be found online at http://dx.doi.org/10.7717/peerj.2935#supplemental-information.

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
