# Peer review of "Evidence for high inter-generational individual quality in yellow-eyed penguins"

_PeerJ, doi:10.7717/peerj.2935_

## Round 0.1 · original submission · Major Revisions

· Academic Editor

Major Revisions

Dear Dr van Heezik,

Thank you for your submission, which has now been seen by three reviewers. As you will see, all are positive, but two of our reviewers raise very similar concerns about your paper, especially with respect to potential collinearity among your variables and the fact that you do not consider possible cohort effects. I would therefore be very grateful of you could consider these comments in detail and revise your paper in the light of them.

I look forward to seeing a new version soon,

all the best,
Lou

Reviewer 1 ·

Basic reporting

Please see general comments.

Experimental design

Please see general comments.

Validity of the findings

Please see general comments.

Comments for the author

Summary: Stein et al. make use of a long-term (23 yr) banding study of yellow-eyed penguins (an endangered species for which the causes of decline are still somewhat uncertain) to classify individuals in terms of breeder quality. The authors link breeding quality to several life-history characteristics of the individuals studied. The major finding of the study is that fewer than 5% of the individuals tracked produce “grand-offspring” and hence that a small proportion of the population contributes to successive generations. The authors argue that this information could be used to focus conservation efforts on highly successful breeders.

Review Summary: An interesting study on a novel topic for the species. The manuscript is also well written for the most part, but could be cleaned up some for clarity and readability. More importantly, there are several issues with the analysis that should be addressed prior to publication. The three biggest issues are the lack of consideration of: 1) cohort/banding year as a driver of lifetime reproductive success, 2) the mates of individuals studied, and 3) collinearity in descriptor variables. Furthermore, with the information provided, the link between the study’s findings and conservation guidance is tenuous. If the authors wish to retain this link in the manuscript, they should spend more time exploring it – or alternatively, the importance of this link should be mentioned only briefly, particularly given that PeerJ does not use subjective determination of “impact” in their assessments of manuscripts.

Major Comments:
1) The largest issue with the study is the lack of consideration of cohort/banding year, particularly given that the most important covariate in the study was lifespan. The authors note that birds sighted after 2007 may have not completed their breeding lifetime during the study period and hence were not included- I am assuming this means that the authors believe that all included birds have completed their breeding lifetime. However, unless there is some real temporal trend in survival/breeding productivity, this does not seem to be the case after some exploratory analyses with the data provided. Two examples: 1) birds classified as super-breeders are not seen after the 1994 cohort, whereas average breeders are seen in all cohorts 2) the average number of mating attempts for super-breeders is more than double that of average breeders. This is particularly alarming due to the widely-recognized influence of age on breeding success in penguins and other seabird species – later cohorts (and their offspring) may seem less productive because they have not yet hit their reproductive prime. Additionally, some cohorts (e.g. 1988) seem to disproportionately produce super-breeders. If this is a true trend, e.g. due to changing environmental conditions, this is an interesting finding in itself. However, this is not currently explored or discussed in the manuscript. If cohort winds up being the major factor controlling LRS (i.e. if an individual has been around longer, it has bred more), the findings hold considerably less weight. The authors should consider how to best deal with this issue – e.g. are the life history characteristics examined significant when including cohort? (see major comment #3), is the variation in LRS within cohort similar to that across cohorts?
2) The second gap in the study is the lack of discussion of the quality of the mates of the birds included. Is the assumption that super-breeders also tend to mate with other super-breeders? Are there any cases in which two birds that were mated are both considered in the study, and what implications does this have for sample independence? Similarly, are banded chicks of the individuals included in the study also included if they fit the study criteria?
3) The authors briefly mention that the response and predictor variables were non-independent and may have biased results but do not discuss this further. Multicollinearity is a major concern in any regression analysis and should be examined in more detail so that the reader can assess the findings in the context of all pertinent information. How strong exactly is the relationship between covariates? If this relationship is strong, the details of the model simplification routine should be discussed further as the order in which variables are removed becomes important. There are also several ways to test for the influence of covariance on model outcomes, such as the calculation of a variation inflation factor – simple to do in R. As noted above, this extends to some of the covariates not examined, particularly cohort. Are the covariates examined significant after taking cohort into account?
4) The study would be strengthened by less discussion of the survival/recruitment results (e.g. Lines 305-313), which are not the study’s focus, and more analysis of the variation in breeder quality. For example - do super-breeders also tend to do well in “bad years”, or years in which overall colony reproductive success is low? Or is there a limitation to the benefits of being a super-breeder? Another interesting line of inquiry would be whether super-breeders are more or less likely to skip a year of breeding than are average breeders, as this is a topic brought up in the introduction.
5) Due to the high site fidelity of the species it is possible that nest quality could be a factor of interest, as it has been found to impact reproductive success in other penguin species. If it is possible to explore with the database, it might have interesting conservation implications in terms of areas to focus effort.
6) The authors do mention that age at first breeding could be used to identify super-breeders, but more thought should be given to the choice of covariates in the context of conservation implications - i.e. are there any covariates that could be examined with the database that would allow for earlier identification of priority birds for conservation action? This is especially true given that several possibilities are listed in the introduction (e.g. see lines 86-93). Fledging weight of the individual comes to mind as a possibility as it could suggest that the individual comes from a high quality lineage (e.g. parents are good foragers) and is more likely to survive to breed and produce high quality offspring that also fledge at a higher than average weight. If the database is large it might be interesting to use additional covariates and run a PCA to identify which characteristics separate individuals most in terms of breeding quality and could be tracked early in an individual’s life. If there are none, the authors might suggest additional measurements that could be added to the long-term study with little effort and potentially benefit conservation.

Minor Comments on Text:
Line 20: Percent that fledged chicks that recruited to the population does not match with percent listed in the Discussion (Line 345- i.e. 47% vs. 57%)
Lines 29-31: The statement that super-breeders “balance” LRS with lifespan (i.e. long life despite high LRS) seems misleading given that the study finds that the most important variable in explaining LRS is lifespan (i.e. high LRS due in part to a long life). This is also true of line 435, where this statement is repeated.
Lines 39-45: Sentence is too cumbersome, particularly to be the jump-off point for the remainder of the manuscript.
Lines 46-49: Are there more recent references you can use here?
Line 77: Reference?
Line 82-83: Could use rewording
Lines 96-101: These two sentences probably belong somewhere in the introduction but seem out of place here.
Line 105: Adding “of individually tracked yellow-eyed penguins” after “a 23 year dataset” would make the transition to the species-specific less abrupt.
Lines 110 and 112: Inconsistency in oxford comma use (or non-use) here and in other places in manuscript.
Line 166: Is it possible that any of these juveniles were prospecting from other sites?
Line 223: Are the predictor variables normally distributed?
Line 229: QAIC of candidate models is never given in results. Please include this information.
Line 255; 271: Here and elsewhere, using fewer semi-colons would improve readability of the manuscript.
Lines 289+ (Discussion, general): The discussion would be easier to read if fewer results/numbers were included.
Lines 314-320: Here and elsewhere, breaking up some of the long sentences would improve manuscript readability.
Lines 324-326: Confusing because in other sections it’s implied that juvenile penguin survival is most definitely lower than earlier periods (i.e. Lines 312-313 and 332-337).
Lines 358-359: Awkward sentence structure
Lines 395-397: Confusing because in other sections (Lines 283-284 and 457-458) it is stated that higher LRS individuals tend to breed one year later on average.
Lines 412-415: Sentence confusing, please clarify
Lines 428-431: Awkward sentence structure
Lines 457-458: The use of age at first breeding as an indicator seems like a stretch given how large the range and variability of this seems to be across individuals. This could also be related to the cohort issue – e.g. if breeding productivity of the colony is decreasing over time, individuals may start to breed earlier to fill this gap (compensatory recruitment; e.g. Votier et al. 2008)
Line 470: Could use a comma in this sentence after “young birds”– several other missing commas or inappropriate placement of commas, please double check.
Lines 472-473: Too abrupt of a jump from an Adélie example to the final conclusion of the study.

Minor Comments on Figures and Tables:
Table 3: Is this SE from a within model result or is this SE across models from model-averaging?
Tables 3 and 4: Please make differences between Table 3 model and Table 4 model more apparent in the table titles.
Figure 1: Can you include latitude/longitude and a scale bar?
Figure 4: It’s very difficult to tell the symbols apart, if possible the use of color would help. If not, please use larger symbols and/or symbols that are easier to differentiate. Furthermore, the symbols do not match those in the figure legend. The x axis and y axis labels should also be enlarged.
New Figure Request: It would be good to include a figure showing the breakdown of super-breeders and average breeders by cohort.

·

Basic reporting

The paper is well written, with the applicable literature well researched. Analyses are sound and thus in summary the paper provides a valuable contribution to the demographic literature, and more specifically the literature dealing with penguins.

Experimental design

Exemplary

Validity of the findings

This is a great study, long in the making. It is great that this research team has followed through in the consistent assembly of data for such a long time.

Reviewer 3 ·

Basic reporting

The symbols used in Figure 4 have not been reproduced correctly in the figure legend, such that it is currently not possible to accurately interpret the figure.

Experimental design

No comments.

Validity of the findings

The focus of the study is rather specific, analysing data from a single population and with little discussion of the broader relevance of the study. The main conclusions are not especially novel, concluding that lifespan and age at first breeding are the main predictors of LRS. However, the study makes nice use of a longitudinal dataset to give better insight compared with cross-sectional studies, and so long as comments (see below) are appropriately addressed, the study could have value for conservation application to this endangered species. Some further discussion of the relevance and applicability of the results to other populations would broaden the potential value and impact of the study.

I am concerned that the statistical analysis of LRS considers many different variables, many of which are likely to be correlated in some ways and so the model violates assumptions of non-independence. The potential for bias is acknowledged by the authors (line 224-225), but it is not clear if the authors have taken any steps to attempt to minimise bias besides the use of information theory. While the use of an information-theoretic approach is, in general, an appropriate choice when there are many possible independent variables, it can also yield biased estimates, particularly when predictors are highly correlated but have very different effects on the dependent variable (see for example Freckleton 2011, though there are many other relevant pieces of literature). I would advise the authors to first investigate extent of collinearity between predictors and use this to inform the selection of models to test.

Also, the statistical approach does not account for non-independence of mates. I suggest that a random factor for “breeding pair” should be included. It is perhaps also relevant to consider the potential for related individuals to not be completely independent of one another.

The study does not take into consideration the possibility of cohort effects, which are widely documented in birds and other animals. In terms of management of the population (presented as a main objective for the study), it would be very useful to understand if there are strong cohort effects. For example, if those born in years of especially favourable conditions (e.g. mild climate, good foraging conditions, high prey availability) are especially high quality individuals, it may be possible to determine the conditions that give rise to “super breeders” and enable detection of those high quality birds early in life. Cohort effects could be incorporated into models by including a variable of year of birth. If cohort effects are found, a priority for further studies should be to try to understand what underlies those cohort effects, which can be used to inform appropriate conservation strategies.

There does not appear to be a statement regarding the availability of the data.

Comments for the author

I think the authors need to review their use of terminology in respect of ‘fecundity’. As far as I can see, fecundity is not actually quantified in the context of the study, but the authors talk about “highly fecund” individuals. Fecundity should not be confused with LRS, as it refers to the reproductive rate of an individual or population. The authors conclude that the “highly fecund” birds had longer lifespans, which explained their higher LRS, relative to “ordinary” birds. This in fact suggests that both groups of birds could have the same fecundity, i.e. same reproductive rate, but those that live longer have more opportunities to breed and thus higher LRS. It therefore seems inappropriate to refer to birds as “highly fecund” individuals.

Line 14-16: this line is incorrect. It should end after “…which in turn raised offspring”, as the study only considers the breeding success of generations 0 and 1, not of the second generation.

Lines 139-141: It is not clear what is the link between yellow-eyed penguins being inshore foragers and the likely for bias due to flipper bands being minimal.
Moreover, if there is the possibility for bias (even if small) due to flipper bands, presumably this is easy to check for or account for in the analysis.

Sentences starting at line 167 and 169 seem to be contradictory. The first says that there is sufficient data from birds that survived to 2 years old, but didn’t necessarily breed (“were potential breeders”), while the next sentence implies there is only data from birds that did attempt to breed.

Lines 177-182: This perhaps belongs in the results section.

Line 185: Regarding the reference to “multiple visits”: presumably a fixed threshold was applied to this?

Line 189-191: It is not clear that the first part of this sentence – concerning birds marked elsewhere - is relevant, since it was stated at line 148 that the analysis only uses data from birds marked at Boulder Beach. It is also not clear why birds found dead more than five years after marking were removed from the analysis; this suggests the possibility for biasing the calculation of number of birds that recruited to the natal site.

Line 192-193: This information has already been said at line 148-149.

line 196: Are the parameters relating to recruitment simple binary (0/1, yes/no) variables? Clarify in the text.

Line 198: This is the first mention that birds were captured again after initial banding as a chick or juvenile. Earlier in the methods, there has only been mention of “resightings”. Mention earlier in the methods about the extent of data originating from (re)captures and sightings. Furthermore, perhaps these lines (198-203) should come earlier (as it relates to individuals/data included in analysis) and could be said more succinctly, e.g. “only birds of known sex (derived from measurements or assumed if mate of known sex) were included in the analysis”.

Line 203-205: This is results. Also, for clarity, it is probably clearer to remove birds of unknown sex from these figures and only cite figures for sample of birds of known sex.

Line 222: Specify the life-history traits that were included in the analysis. Presumably the same as used in the Wilcoxon rank-sum tests, explained in the paragraph above, but needs to be clarified.

Line 232: The study only considers reproductive success over two generations: generations 0 and 1.

Line 236: Again, need to state what life-history parameters were considered.

Line 268-269: Need to quote relevant statistics (not pseudo R-squared) for both lifespan and age at first breeding.

Line 270: This contradicts with the result presented at line 255.

Line 274: There is no measure of fecundity presented, so there can be no claims about birds with above-average fecundity.

Paragraph starting at line 327: There are lots of possibilities put forward that could contribute to high juvenile mortality. What is not clear is if there is any evidence for any of these mechanisms within the study population.

Line 385: Unclear what is meant here by fitness. Darwinian? Health?

---

## Round 0.2 · Minor Revisions

· Academic Editor

Minor Revisions

Dear Yolanda

Thank you for your very comprehensive revision, and for dealing with the reviewers' comments in such a positive way. Having gone over your revision, and response, I am now happy to accept your paper for publication, but suggest two really very minor additions/changes.

1. A highly pedantic point!! In the abstract, you say 'c. 24.6%'. Given you've added the "circa" there, it might be more appropriate simply to say "c.25%"

2. Given the constructive suggestions of the reviewers, and the obvious effort they put into reviewing your MS, I just think it would be nice to thank them in the acknowledgments. Peer review can so often be a thankless task, that any small gesture of appreciation can be very welcome!

with best wishes,
Lou

---

## Round 0.3 · accepted · Accept

· Academic Editor

Accept

Hello again Yolanda

Many thanks for sorting that out, and congratulations on a fine piece of work. I hope you can now have a rest, and a very merry Christmas!

All the very best for 2017,

Cheers,
Lou